# Photothermal Thin Films with Highly Efficient NIR Conversion for Miniaturized Liquid-Crystal Elastomer Actuators

**DOI:** 10.3390/polym14152997

**Published:** 2022-07-24

**Authors:** Wei-Yi Wang, Bo-You Lin, Yen-Peng Liao, Yao-Joe Yang

**Affiliations:** Department of Mechanical Engineering, National Taiwan University, Taipei 10617, Taiwan; falinewang@mems.me.ntu.edu.tw (W.-Y.W.); lby97911@mems.me.ntu.edu.tw (B.-Y.L.); liaoyenpeng@mems.me.ntu.edu.tw (Y.-P.L.)

**Keywords:** photothermal film, liquid metal, shape-memory polymer, liquid-crystal elastomer, NIR-driven actuator, rectilinear locomotion

## Abstract

This work presents the development of highly efficient photothermal thin films (PTFs) and the demonstration of their application on miniaturized polymer-based soft actuators. The proposed PTF, which comprises acrylic-based black paint and EGaIn liquid metal (LM) microdroplets, serves as an excellent absorber for efficiently converting near-infrared (NIR) irradiation into heat for actuating liquid-crystal elastomer (LCE) actuators. The introduction of LM microdroplets into the PTFs effectively increases the overall thermal efficiency of PTFs. Miniaturized soft crawlers monolithically integrated with the NIR-driven LCE actuators are also implemented for demonstrating the application of the proposed PTF. The crawler’s locomotion, which is inspired by the rectilinear movement of snakes, is generated with the proposed PTF for inducing the LC-to-isotropic phase transition of the LCEs. The experimental results show that introducing LM microdroplets into the PTF can effectively reduce the thermal time constants of LCE actuators by 70%. Under periodic on/off NIR illumination cycles, the locomotion of crawlers with different dimensions is also demonstrated. The measurement results indicate that the proposed PTF is not only essential for enabling photothermal LCE actuation but also quite efficient and durable for repeated operation.

## 1. Introduction

In recent years, photo-responsive materials, which convert light energy directly into mechanical work, have attracted significant attention because of their promising applications in light-driven mechanical actuation. Liquid-crystal elastomers (LCEs) have been demonstrated to be some of the most excellent materials for light-driven actuation [1,2,3,4,5,6]. LCEs exhibit large and reversible deformations due to the liquid-crystal phase transition induced by heating and cooling cycles [7,8,9,10,11,12].

The typical actuation schemes of LCEs, such as electrothermal actuation and photothermal actuation, originate from the characteristics of rubber elasticity and the orientational order of liquid crystals. In general, LCEs require external heating elements to facilitate shape-change actuation. Previous research reported LCE composites that were electrically actuated by Joule heating using conductive paths such as flexible metal wires [13,14,15,16], silver ink/paste [17], carbon coatings [18], and encapsulated liquid metal [19,20]. These approaches have proven to be effective for repeatable actuation. However, the physical and mechanical constraints due to the electrical conduction path to the heating elements limit the operation and deployment of the devices [21].

Therefore, in recent years, photothermal conversion induced by visible or near-infrared (NIR) light [22,23,24,25,26,27] has been regarded as a promising actuating scheme for LCEs, since the deformation of LCEs can be actuated in a contactless way. Tian et al. presented a NIR-driven, polydopamine-coated LCE that can serve as an optically driven artificial muscle [28]. The proposed device is capable of bending or rolling up by the surface scanning of a NIR laser. A NIR-induced, shape-morphing LCE that exhibits excellent repeated and fast actuation was reported [29]. The device is capable of maintaining an actuation strain of 56% and can be morphed into various shapes by using different masks for actuation illumination.

It has been reported that rigid nanoparticles, such as gold nanospheres/nanorods and carbon nanotubes (CNTs), can serve as the heating fillers for LCEs. In [26], light-driven LCE microactuators incorporating gold nanospheres and nanorods were proposed. Reversible actuation with a large-strain response was also observed and reported. Wu et al. presented light-induced nanocomposites consisting of LCEs and single-wall CNTs (SWCNTs) with multimode NIR photomechanical actuations [30]. The allyl sulfide-based composite exhibits a strong photothermal effect and demonstrates various actuations, such as contraction, bending, and curling. Additionally, Wang et al. reported an adaptive soft robot that comprises bimorph LCE structures, actuators of thin resistive heaters, and sensors of Si-based photodetectors, for mimicking the locomotion of a caterpillar [31]. Kohlmeyer et al. reported wavelength-selective, IR light-driven LCE bilayer hinges that exhibit fast and reversible bending with a large strain. Various remote-controlled soft actuators, including active origami structures, Venus flytrap-inspired grippers, and climbing inchworm walkers, have been demonstrated using the proposed LCE structures [32]. In [33], the development of a thermomechanical LCE actuator containing gold nanoparticles (AuNPs) for improving material response time was reported. Under fast heating conditions, the proposed LCE actuators exhibited an obvious increase in the rate of change in strain with respect to time. Li et al. presented single-wall CNT-incorporated LCE nanocomposites that exhibit reversible photoactuation capability [23]. The transition temperatures of the SWCNT–LCE nanocomposites were found to be significantly lower than those of the blank LCE.

In this work, we present the development of highly efficient photothermal thin films (PTFs) and demonstrate their application on miniaturized LCE-based crawling devices. The PTF comprises acrylic-based black paint and EGaIn liquid metal (LM) microdroplets, and the miniaturized soft crawling device comprises a light-driven arched LCE structure spray-coated with PTF. The primary purpose of the PTF is to efficiently absorb and transform NIR light into thermal energy to effectively induce the actuation of the LCE-based crawling devices. Additionally, the introduction of LM droplets into the PTF intends to increase actuation efficiency without the degradation of the mechanical properties of the actuators. This paper is organized as follows: Section 2 describes the design and operation principles of the PTF and the NIR-actuated LCE crawling device and introduces the fabrication process of the devices in detail; Section 3 describes the experimental results and highlights the effects of the PTFs on the LCE actuation; finally, Section 4 presents our conclusions.

## 2. Materials and Methods

### 2.1. Design Strategy of Photothermal Actuation

Figure 1 shows the schematic of the light-driven LCE crawling device with the proposed PTF. A device, which consists of two claw-like legs and an arched LCE actuator connecting the legs, was designed for demonstrating the LCE actuation with the PTF. The structure of the device is monolithically made of LCEs. By employing the shape-programming process, the LCE structure is capable of reversibly exhibiting large deformations between two stable states induced by the LC-to-isotropic phase transition. As shown in the figure, a PTF layer, which is an excellent solar absorber for efficiently converting NIR light into heat, is selectively coated on the top of the arched LCE actuator. The PTF comprises commercially available black acrylic paint (BAP) and EGaIn LM microdroplets. The BAP is primarily made of three ingredients: water, acrylic resin, and black pigment, and it serves as the matrix of the PTF, which can be easily deposited on LCE surfaces using a spray-coating process.

Under NIR illumination, the PTF of the arched actuator absorbs light energy and heats up the arched LCE structure. As the temperature rises above the LC-to-isotropic transition temperature, the actuator extends. As NIR illumination is terminated, the actuator temperature decreases gradually, and the actuator starts to retreat to its original shape. Under the periodic on/off application of NIR illumination, the device simulates the rectilinear locomotion in succession. The details of the locomotion principle can be found in the Appendix A. The proposed remote actuation method of the LCE by using NIR light illumination is the key to realizing untethered devices and possesses several advantages, such as simple configuration, remote actuation, and adaptability in harsh environments. In addition, because the thermal conductivity of LM is relatively high, introducing microscale LM droplets (EGaIn) into the PTF can effectively enhance the thermal efficiency of the NIR-driven actuator. Additionally, LM exhibits extremely low stiffness due to its liquid nature, and, thus, it can freely deform with the surrounding matrix with little thermal resistance [34]. Therefore, by mixing LM into the PTF, it is possible to increase the overall thermal conductivity of the PTF while maintaining its compliance and conformity. By increasing the thermal conductivity, the thermal time constant of the heat-induced actuator decreases, and, thus, the thermal performance of the actuator could be improved. Additionally, the huge NIR absorption difference between the PTF and the LCE structure allows the LC-to-isotropic phase change to selectively occur in the specific region under universal illumination. Therefore, it is possible to achieve a more accurate and customized means of actuation by designing the shape and location of the PTF.

### 2.2. Fabrication

Figure 2 describes the process flow for fabricating the LCE crawling devices with PTFs. The LCE structure was fabricated using a lithography-based molding process. First, to prepare the LCE prepolymer solution, LC mesogens (RM257, Wilshire Technologies, Princeton, NJ, USA, 95%, 1.50 g) were dissolved in toluene (0.45 g) at 85 °C followed by cooling down to room temperature. Then, a chain extender (EDDET, Sigma-Aldrich, San Luis, MO, USA, 95%, 0.317 g), photoinitiator (HHMP, Sigma-Aldrich, San Luis, MO, USA, 98%, 0.009 g), catalyst (DPA, Alfa Aesar, Haverhill, MA, USA, 99%, 4.18 mg), and tetra-functional thiol cross-linker (PETMP, Sigma-Aldrich, San Luis, MO, USA, 95%, 0.141 g) were added to the solution to form loosely cross-linked LCE networks via the thiol–Michael addition reaction [35]. After being thoroughly mixed and degassed, the prepared LCE prepolymer solution was blade-coated in a PMMA mold, as shown in Figure 2a. The prepolymer was placed at room temperature for 8 h. Subsequently, the prepolymer was cured at 85 °C for over 24 h for solvent evaporation (Figure 2b). Then, as shown in Figure 2c, the mold was removed. Figure 2d shows that a shadow mask was aligned to pattern the PTF layers. Figure 2e illustrates that the PTF layer was deposited on the LCE by spray-coating BAP (Black 2.0 Paint^®^). Note that, to improve PTF thermal conductivity, LM (eutectic gallium–indium droplets) was also dispersed into the BAP before depositing the BAP on the LCE structure. LM microparticles (5 g) were formed in the BAP (1 g) and DI water (0.5 g) through agitation with a sonicator (Qsonica, Newtown, CT, USA, Q700) for 5 min. The sonicator was programmed to pause for 2 s between every 4 s of sonication during the operation. Figure 2f shows the removal of the mask.

Figure 3 illustrates that the shape programming of the LCE was realized by using a method combining mechanical stretching and thermal cross-linking. The mechanical stretching process is equivalent to an embossing technique that aligns the LC molecules with a specially designed PMMA mold. The mold, which has specially designed features of shapes, was pressed against the LCE structure (Figure 3a) and then cured for 24 h under a 365 nm UV light (Figure 3b) to program permanently aligned monodomain LCE samples for achieving shape deformation under external stimuli. This process changed the shape of the previous structure (as shown in Figure 2f) into the shape defined by the PMMA mold. The embossing process provides several advantages, such as producing three-dimensional structures, a relatively low cost, simple operation, and high replication accuracy for small features. Finally, the fabricated device was removed from the mold, as shown in Figure 3c.

Figure 4a shows a fabricated LCE device. The length, height, and thickness of the device are 10 mm, 6 mm, and 3 mm, respectively. Figure 4b is the SEM image of the PTF deposited on the LCE structure.

## 3. Results and Discussion

### 3.1. Influences of PTFs on LCE Thermal Behaviors

Figure 5a-i presents the transient responses of temperature under NIR illumination (808 nm, 2.4 W/cm^2^) for the LCEs of different configurations, as described in Table 1 (i.e., Type-1, Type-2, Type-3, and Type-4). Figure 5a-ii uses the same data set as Figure 5a-i but has a different temperature scale for illustrating the details of curves of Type-3 and Type-4. In order to quantitatively compare the heating performance among these devices, NIR illumination was vertically incident upon the central top surface of the arched actuator of each device. The temperature of the arched actuator under NIR illumination was measured by using a non-contact infrared thermometer (Optris, Berlin, Germany, LaserSight). The thermometer, which was placed beside the NIR light source, pointed to the top surface of the arched actuator to measure the temperature. Note that the infrared thermometer has a spectral response of 8–14 μm, and, therefore, it will not interfere with the NIR light source of 808 nm in wavelength. Figure 5b-i show the close-up region of the temperature increase from 0 s to 20 s. Figure 5b-i, b-ii show the same curves with different temperature scales. Each data point in the figure is the average value of the measured temperatures obtained from five heating cycles, and the error bars of each point are the measured minimum and maximum values. Under the same NIR illumination for 120 s, the temperatures of the LCEs with PTFs (Type-1 and Type-2) were elevated to 160 °C, while the temperatures of the LCEs without PTFs (Type-3 and Type-4) were around 28 °C. When the NIR was switched off, the devices started to cool down. The corresponding cooling process of the devices is also shown in this figure. These results indicate that the photothermal conversion efficiency of the PTF is sufficient to trigger the LC-to-isotropic phase transition for inducing LCE deformation, which occurs at around 80 °C [35]. The figure also shows that the heating (cooling) time of Type-2 to achieve the LC-to-isotropic (isotropic-to-LC) phase-change temperature was about 70% less than that of Type-1, which indicates that adding LM can effectively improve the thermal responses of the device. Comparing the thermal responses of these devices, the device coated with a thin BAP mixed with LM (i.e., Type-2) exhibited the best performance. Therefore, based on the same configuration, we implemented three crawling devices of different sizes and studied their thermal performance and locomotion.

Figure 6 provides pictures of these crawling devices (i.e., Device-A, Device-B, and Device-C). Table 2 lists the dimensions of these devices. Note that the lengths and heights of these devices are proportionally scaled, while their thicknesses are the same. Figure 7a shows the thermal responses of Device-A, Device-B, and Device-C heated by NIR illumination (808 nm) with 2.4 W/cm^2^. Similar to the configuration shown in Figure 5, NIR illumination is vertically incident upon the central top surface of the arched actuator of each device, and each data point is the average of five measured temperature values. The error bars indicate the minimum and maximum values. The devices were heated for 120 s and then passively cooled down to room temperature. The temperatures of these devices reached a steady-state well before the heating was turned off. In addition, smaller devices have higher steady-state temperatures. This is because the acquired heat is proportional to the surface, while the heat capacity is proportional to the volume. Furthermore, Figure 7a clearly indicates that 120 s of illumination was sufficient for all the devices to reach the LCE phase-change temperature (i.e., above 80 °C) for device actuation. Therefore, it was possible to optimize (i.e., shorten) the heating time and speed up the actuation of the devices. Figure 7b shows a close-up view of Figure 7a for the duration from 0 s to 20 s. The dotted line in the figure indicates the temperature that is sufficient to achieve LCE phase change. Therefore, the intersection points of the dotted line and the transient temperature curves indicate the required heating times for initializing LCE phase change were about 8.2 s for Device-A, 2.2 s for Device-B, and 1.9 s for Device-C. These results represent the minimum values of heating times for these devices. Based on our preliminary measurement results, the optimized reliable heating times for Device-A, Device-B, and Device-C were 15 s, 6 s, and 5 s, respectively. By using these heating times, the measured transient temperature behaviors with the same NIR illumination (808 nm and 2.4 W/cm^2^) are shown in Figure 8. The temperatures of Device-A, Device-B, and Device-C quickly reached 120 °C, which is well above the temperature required for device actuation (i.e., LC-to-isotropic phase-change temperature). The NIR actuation schemes presented in Figure 8 were applied in the following crawling experiments.

### 3.2. Demonstration of LCE Crawling Devices with PTFs

Figure 9 illustrates the experimental setup for measuring the crawling locomotion of the devices. To quantitatively measure the locomotion of the device, the NIR light source was fixed on the translational linear stage above the measurement platform. The light source moved with the crawling device during the measurement, so that the NIR beam was vertically incident on the arched actuator for ensuring that the heating power is the same for each actuation cycle. The locomotion snapshot images of an LCE crawler (Device-B) for an actuation cycle are shown in Figure 10. With the heating by NIR irradiation, the photothermal effect of the PTF flattened the arched actuator and thus stretched the crawler. After turning off the NIR, the device started to recover to its initial shape, and the body moved forward when pulled by the front leg, which hooks on the ground. The displacement of the front leg tip was about 2 mm for each actuation cycle. Note that in the actuation cycle, the front leg may slightly slide backward as the device recovers to its original shape during the cooling process; however, this backward sliding is insignificant compared with the total forward motion distance of each step.

Figure 11 presents the transient displacements of 10 actuation cycles for Device-A, Device-B, and Device-C. The associated curvatures of the devices are also shown in the figures. The accumulated displacement of the front leg tip is the total forward displacement of the device. Under the periodic on/off illumination cycles, the devices constantly deformed between the original state and the flat-shape state and generated rectilinear locomotion, which mimics the rectilinear locomotion of a snake. The reversible shape-morphing process led to the continuous forward movement of the devices. The total crawling displacement for 10 cycles of actuation was 26.2 mm, 20.1 mm, and 18.2 mm for Device-A, Device-B, and Device-C, respectively. For each crawler, the measured displacement achieved by a single actuation cycle was quite consistent for these 10 cycles. The measured transient curvatures also exhibited a similar phenomenon. These results indicated that the crawlers showed good consistency in the reversible deformations. Additionally, the PTFs deposited on these devices exhibited excellent durability after repeated operations.

## 4. Conclusions

In this work, we developed highly efficient PTFs and presented the PTF application on light-driven miniaturized LCE-based actuators. The LCE device, which was coated with the PTF on a specified region, generated a reversible deformation from its original shape to a flat shape under repeated NIR light irradiation. The experiment results showed that the photothermal conversion efficiency of the PTF was sufficient to trigger the LC-to-isotropic phase transition for inducing LCE deformation. The locomotion of LCE crawlers under periodic NIR excitations was successfully demonstrated. In addition, the introduction of liquid metal into the PTF effectively reduced the heating and cooling times by about 70%.

## Figures and Tables

**Figure 1 polymers-14-02997-f001:**
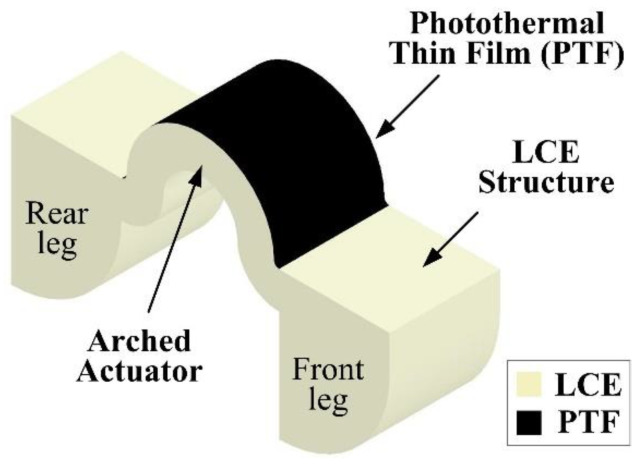
Schematic of a light-driven LCE device with PTF.

**Figure 2 polymers-14-02997-f002:**
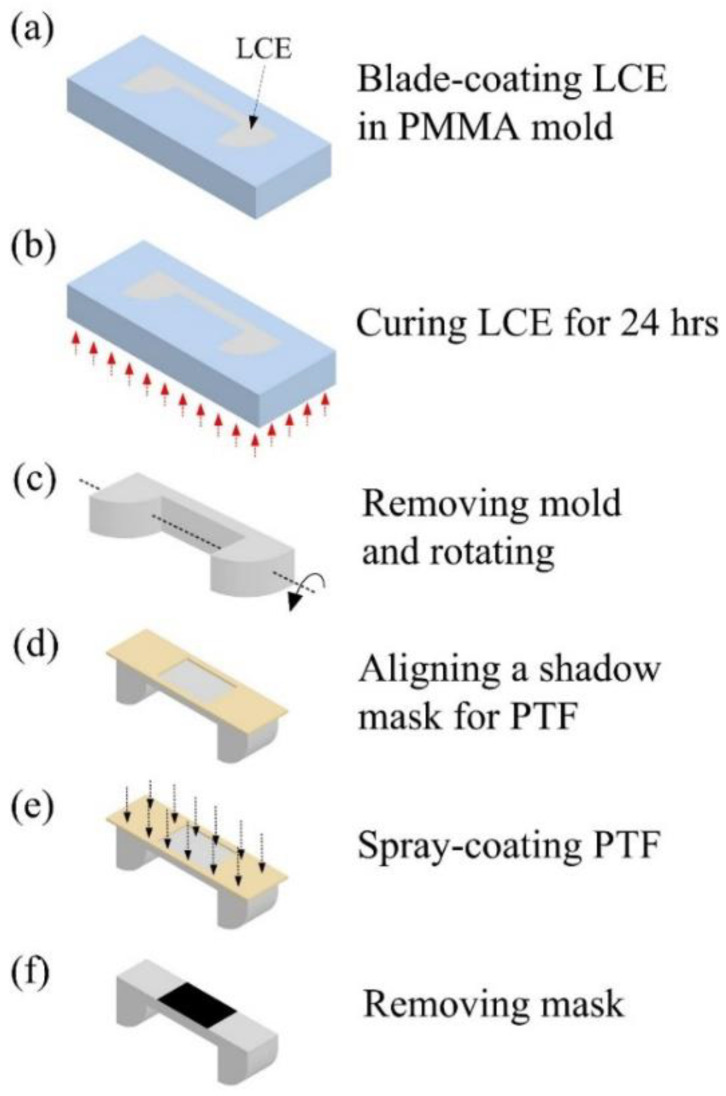
The fabrication process of forming the structure of the light-driven LCE device.

**Figure 3 polymers-14-02997-f003:**
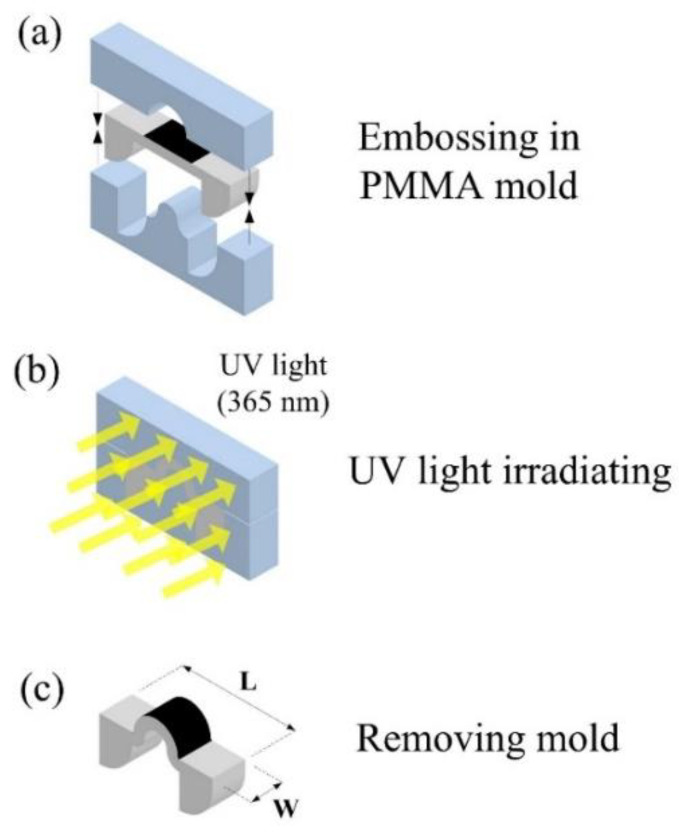
The shape programming process for realizing the arched actuator with PTF.

**Figure 4 polymers-14-02997-f004:**
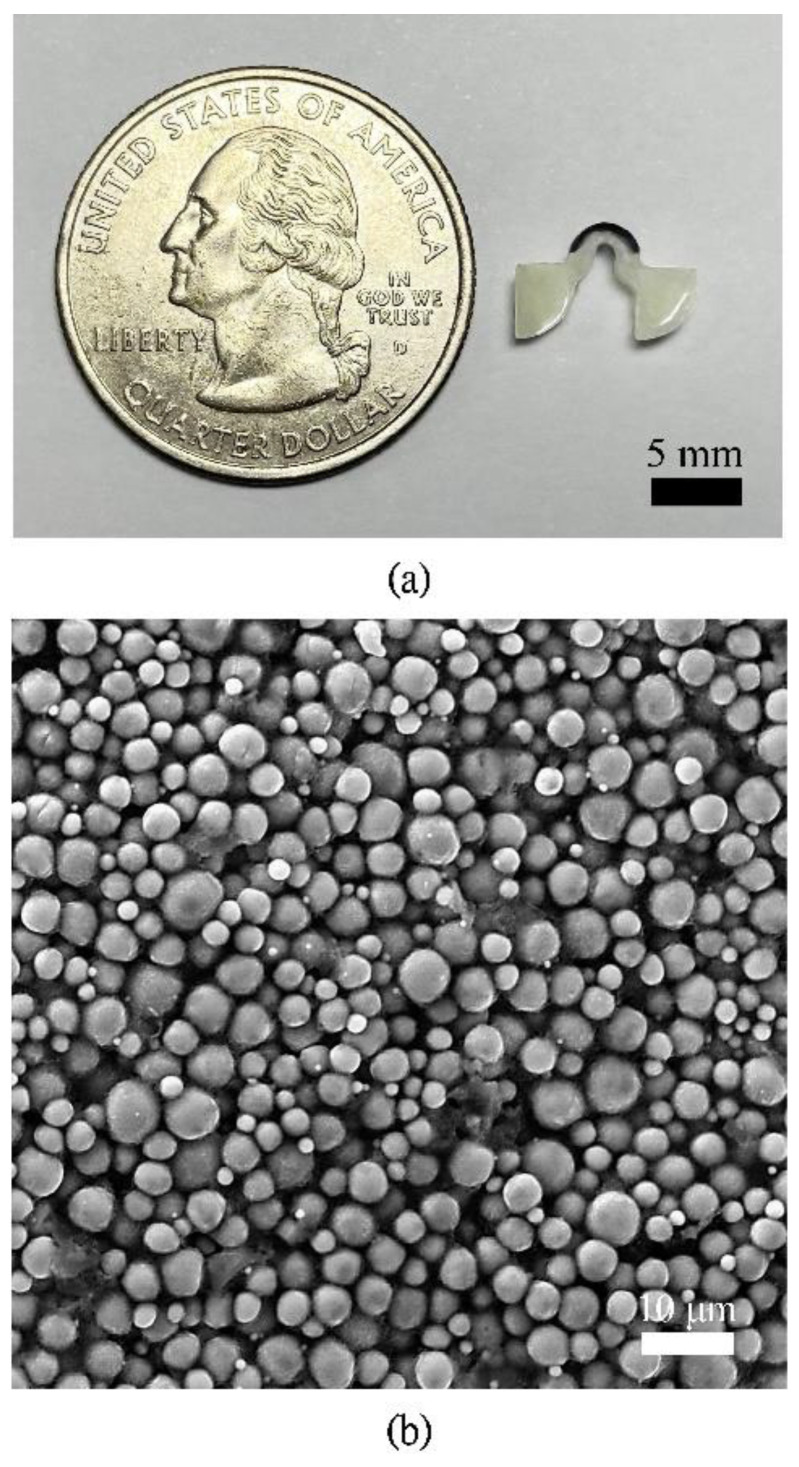
Fabrication results: (**a**) picture of the fabricated LCE device; (**b**) SEM image of the top surface of the LCE device deposited with PTF.

**Figure 5 polymers-14-02997-f005:**
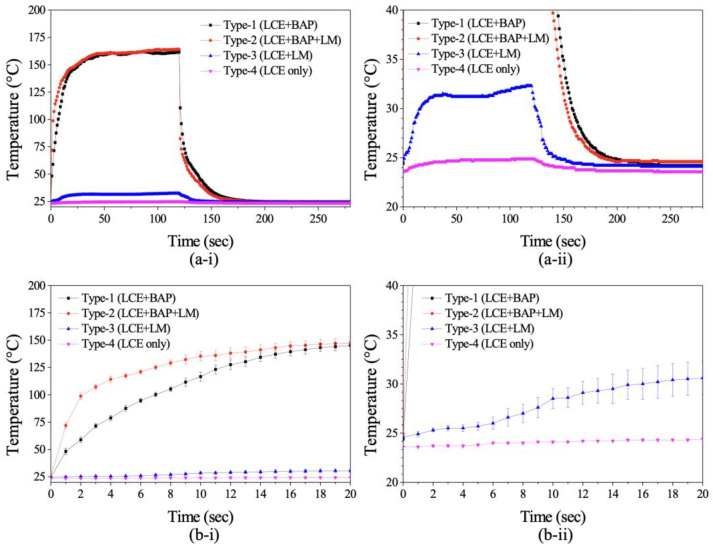
(**a-i**) Transient temperature responses of different types of LCEs heated by NIR light (808 nm, 2.4 W/cm2): (**a-ii**) The close-up figure of (**a-i**); (**b-i**) the close-up of (**a-i**); (**b-ii**) The close-up of (**b-i**).

**Figure 6 polymers-14-02997-f006:**
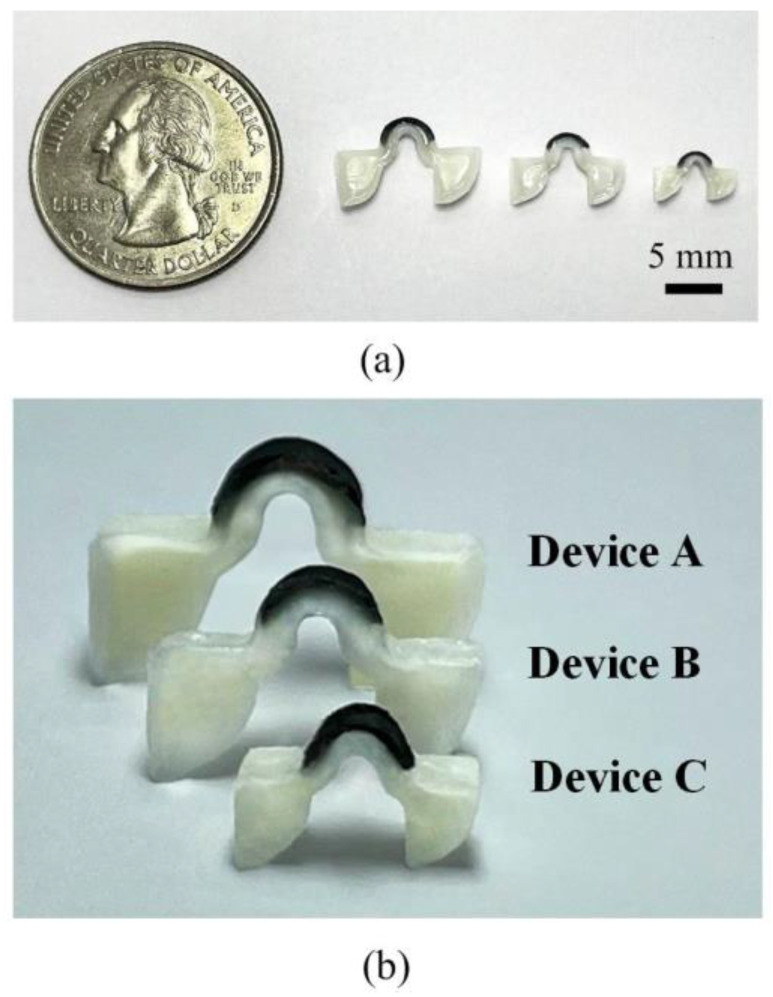
Fabricated LCE devices of different sizes: (**a**) side view of the devices with a scale bar. (**b**) side view of devices at standing position ready for locomotion.

**Figure 7 polymers-14-02997-f007:**
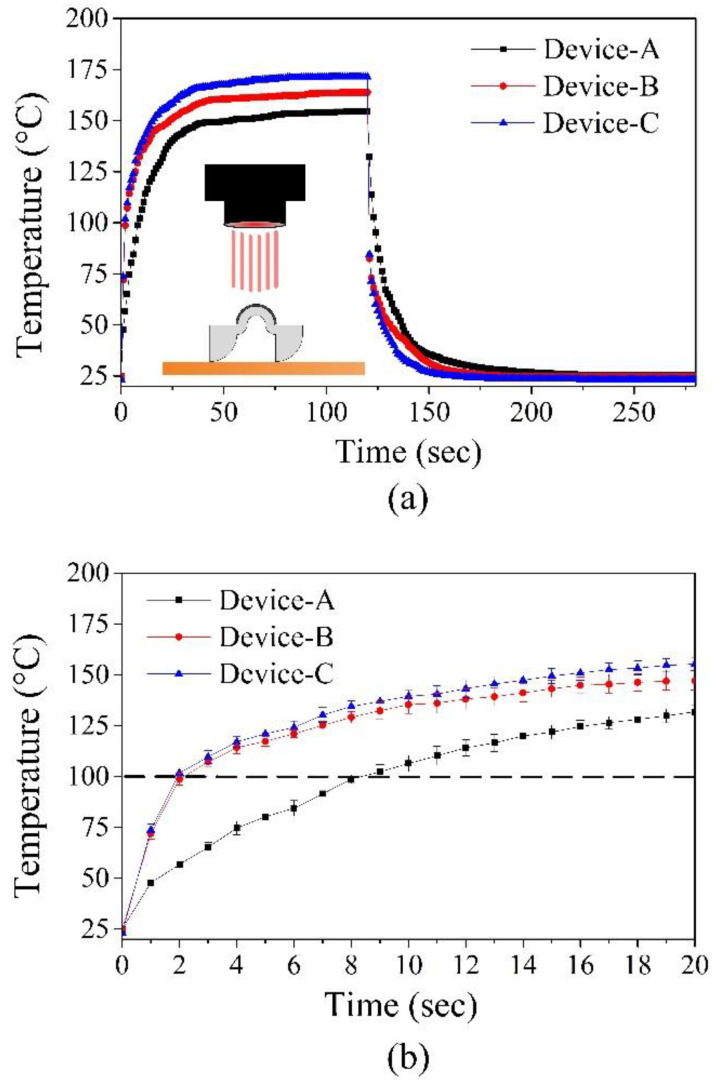
(**a**) Transient temperature responses of Device-A, Device-B, and Device-C under NIR light illumination (808 nm; 2.4 W/cm^2^); (**b**) the close-up of Figure 7a.

**Figure 8 polymers-14-02997-f008:**
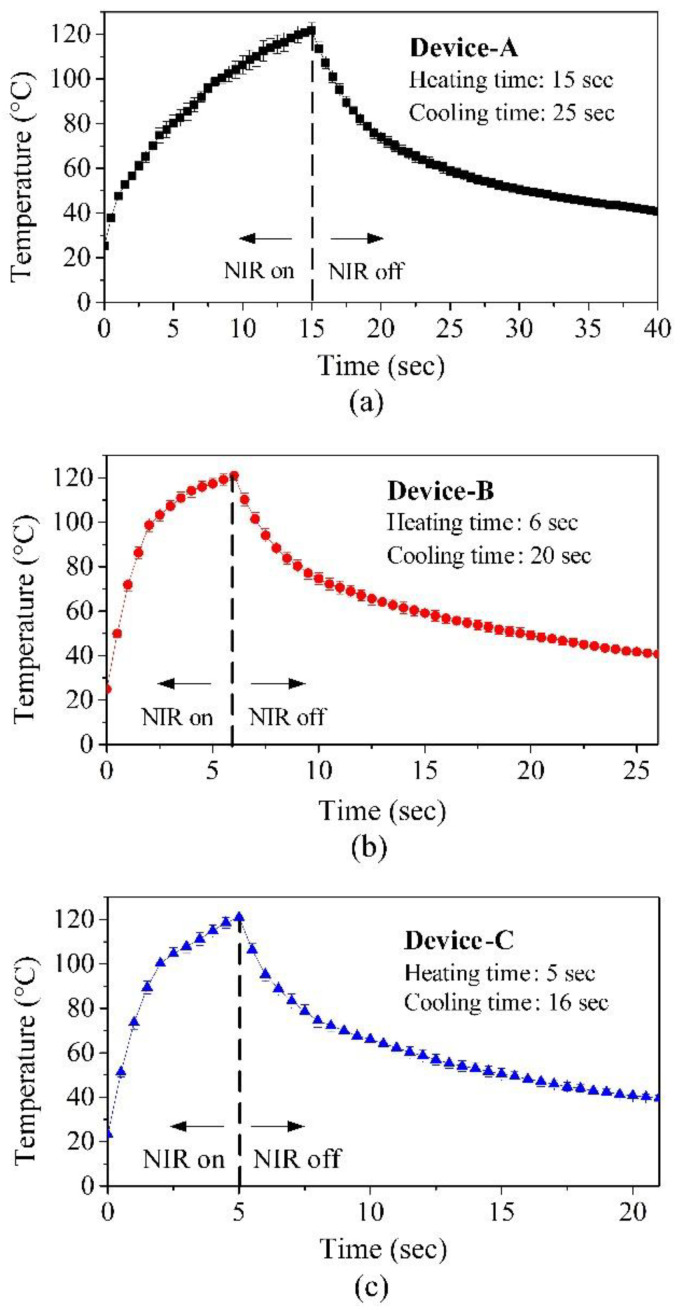
The transient temperature responses of (**a**) Device-A, (**b**) Device-B, and (**c**) Device-C as a function of time with the NIR light illumination (808 nm, 2.4 W/cm^2^).

**Figure 9 polymers-14-02997-f009:**
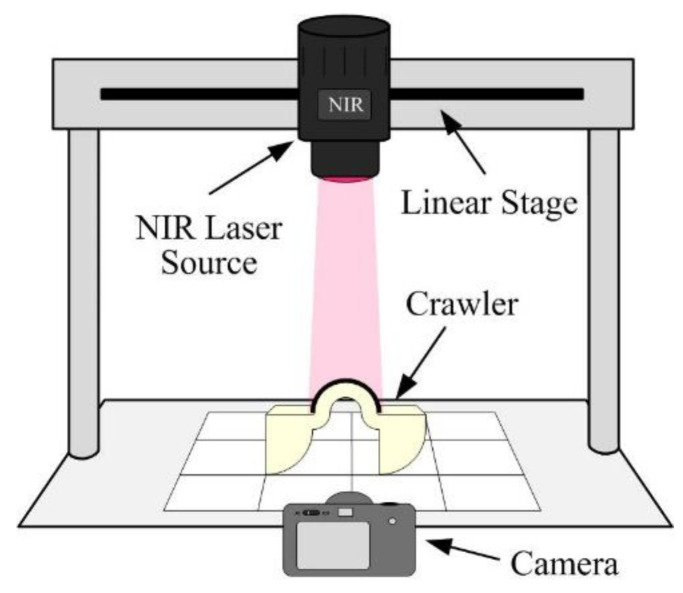
The experimental setup for measuring the crawling locomotion of devices.

**Figure 10 polymers-14-02997-f010:**
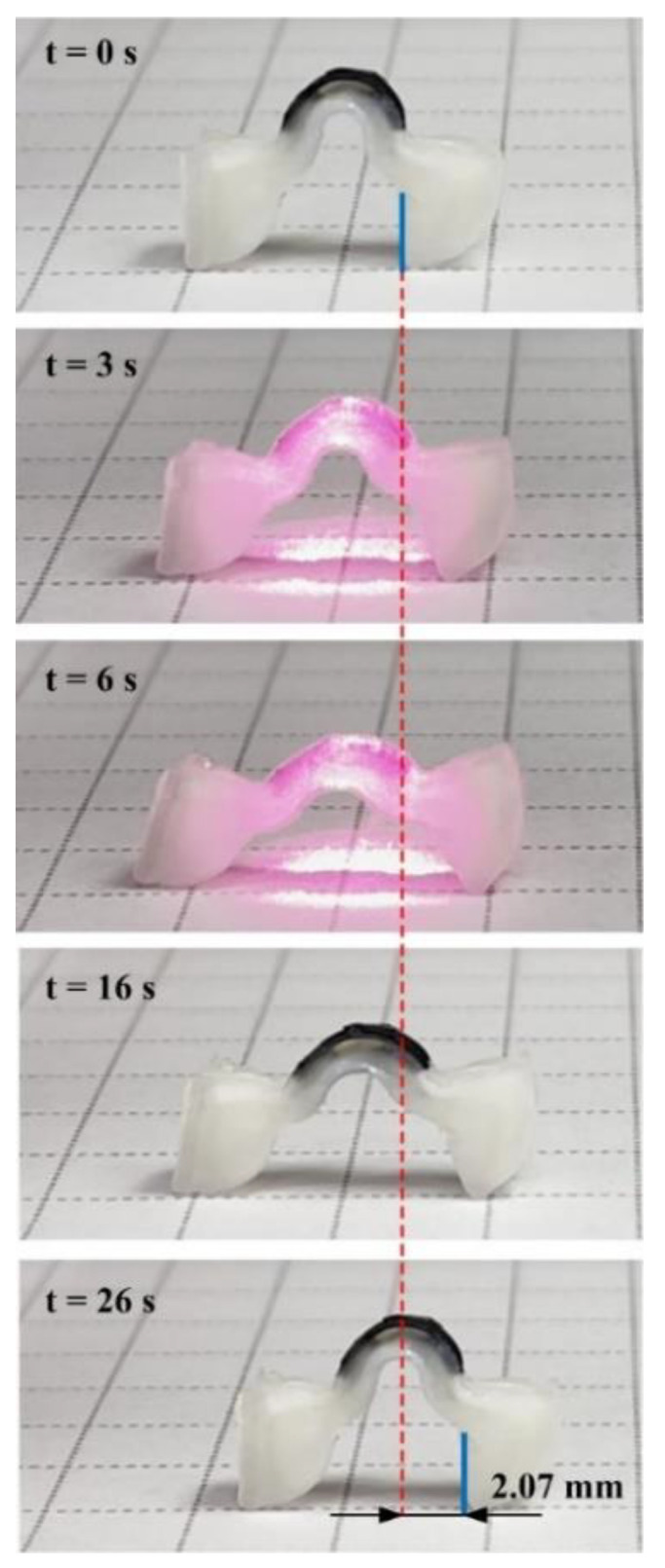
The snapshot images of an LCE crawler (Device-B) for one actuation cycle.

**Figure 11 polymers-14-02997-f011:**
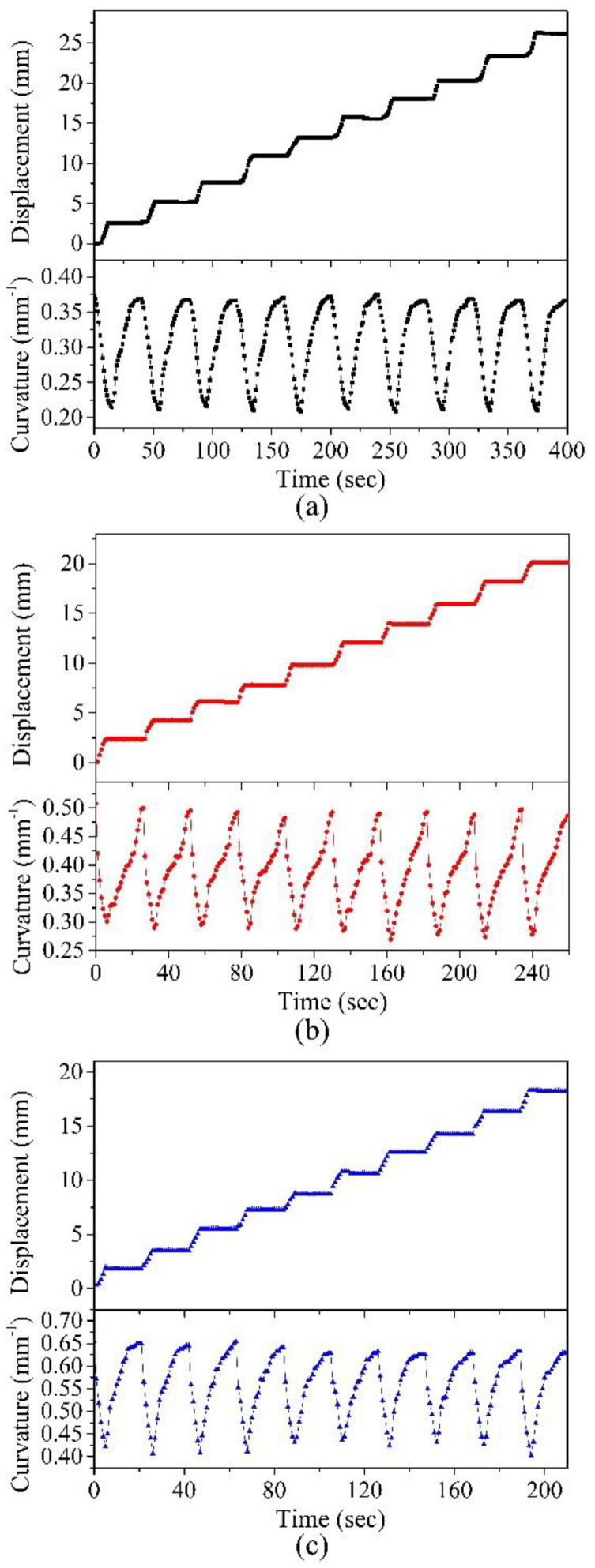
Measured transient displacement and the measured actuators curvature change in (**a**) Device-A, (**b**) Device-B, and (**c**) Device-C for 10 actuation cycles. The NIR light intensity is 2.4 W/cm^2^.

**Table 1 polymers-14-02997-t001:** The configurations of LCE devices with different photothermal films.

Devices	Configurations
Type-1	LCE with BAP
Type-2 Type-3 Type-4	LCE with BAP + LM LCE with LM LCE (only)

**Table 2 polymers-14-02997-t002:** The dimensions of the LCE devices for locomotion measurement.

	Device-A	Device-B	Device-C
Length (L) (mm) Height (H) (mm) Thickness (W) (mm)	13 7.8 3	10 6 3	7 4.2 3

## Data Availability

Data are available on request.

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
