# Peer review of "Photothermal Thin Films with Highly Efficient NIR Conversion for Miniaturized Liquid-Crystal Elastomer Actuators"

_polymers, 2022, doi:10.3390/polym14152997_

Round 1
Reviewer 1 Report
Dear Authors,
I find the article very interesting and believe that the audience will too. The description of the findings is complete and compelling for the most part.
I would propose to define the term SWCNT, by context probably the authors mean soft-crawler carbon nanotube. Defining acronyms dispels ambiguities.
On Fig. 5.a, I would propose using a second scale of the temperature axis on the right hand side of the figure, for types 3 and 4. Thus, if two scales are used in the same figure, please include clarification in the caption, to avoid confusion. Else, use Fig. 5.b to display types 3 and 4 with an appropriate maximum for the temperature axis, so that their details are clear. In their present representation neither Fig. 5.a nor Fig. 5.b are clear enough, unfortunately. By using a more appropriate scale, also the cooling-down rates of structures without BAP will be better appreciated and compared, to those of structures without BAP. Because the structures are passively cooled down their cooling rates could then also be observed.
I propose to elaborate more on the statement of lines 273 and 274 “crawlers show good consistency in the reversible deformations”. The authors could be more specific about the meaning of the statement, particularly the aspect of "consistency".
What is not described at all—at least briefly—is the methodology to measure temperature of the structures. This parameter is relevant because it gives to the reader an idea of the error in the temperature measurements and of course of the methodology itself. The temperature measurement methodology may differ to the one used by other researchers of similar structures.
In the text it is also not mentioned the methodology to describe the curvature measurements. A ketch would be appropriate. Curvature values are plotted in Fig. 11. This is also an important aspect for the reader.
Best of luck with the publication process.
Reviewer 2 Report
The paper entitled "Photothermal Thin-Films with High-Efficient NIR Conversion for Miniaturized Liquid-Crystal-Elastomer Actuators" reports on the design, construction and evaluation of actuation performance of a device based on photothermal thin films embedded with liquid crystal elastomer.
Overall, the manuscript is well written and the experimental results are good correlated. The clarity of the presented ideas and the effort made for the selection of the components of the action device can only be ascertained. Also, the Figures indicating the mechanism of operating or the integrated elements are well explained and realized.
In these experimental conditions, based on the selected elements and temperature range, which would be the closest application of this device?
I recommend the acceptance of this paper in the present form!
